# Evolutionary and Genetic Recombination Analyses of Coxsackievirus A6 Variants Associated with Hand, Foot, and Mouth Disease Outbreaks in Thailand between 2019 and 2022

**DOI:** 10.3390/v15010073

**Published:** 2022-12-27

**Authors:** Jiratchaya Puenpa, Nutsada Saengdao, Nongkanok Khanarat, Sumeth Korkong, Jira Chansaenroj, Ritthideach Yorsaeng, Nasamon Wanlapakorn, Yong Poovorawan

**Affiliations:** 1Center of Excellence in Clinical Virology, Department of Pediatrics, Faculty of Medicine, Chulalongkorn University, Bangkok 10330, Thailand; 2Department of Microbiology, Faculty of Medicine, Siriraj Hospital, Mahidol University, Bangkok 10700, Thailand; 3FRS(T), The Royal Society of Thailand, Sanam Sueapa, Dusit, Bangkok 10300, Thailand

**Keywords:** coxsackievirus A6, hand, foot, and mouth disease, evolution, complete genome sequencing, genetic recombination

## Abstract

Coxsackievirus (CV)-A6 infections cause hand, foot, and mouth disease (HFMD) in children and adults. Despite the serious public health threat presented by CV-A6 infections, our understanding of the mechanisms by which new CV-A6 strains emerge remains limited. This study investigated the molecular epidemiological trends, evolutionary dynamics, and recombination characteristics of CV-A6-associated HFMD in Thailand between 2019 and 2022. In the HFMD patient samples collected during the 4-year study period, we identified enterovirus (EV) RNA in 368 samples (48.7%), of which CV-A6 (23.7%) was the predominant genotype, followed by CV-A4 (6%), EV-A71 (3.7%), and CV-A16 (3.4%). According to the partial viral protein (VP) 1 sequences, all these CV-A6 strains belonged to the D3 clade. Based on the viral-RNA-dependent RNA polymerase (RdRp) gene, four recombinant forms (RFs), RF-A (147, 84.5%), RF-N (11, 6.3%), RF-H (1, 0.6%), and newly RF-Y (15, 8.6%), were identified throughout the study period. Results from the similarity plot and bootscan analyses revealed that the 3D polymerase (3Dpol) region of the D3/RF-Y subclade consists of sequences highly similar to CV-A10. We envisage that the epidemiological and evolutionarily insights presented in this manuscript will contribute to the development of vaccines to prevent the spread of CV-A6 infection.

## 1. Introduction

Since 2008, the coxsackievirus (CV)-A6 strain has become the predominant genotype associated with global outbreaks of atypical hand, foot, and mouth disease (HFMD) [1,2,3,4,5,6]. Recent surveillance data have shown the epidemic potential of CV-A6 worldwide, including in Brazil [7], China [8], France [9], Hong Kong [10], India [11], Japan [12], Turkey [13], Uruguay [14], and Vietnam [15]. Severe clinical manifestations, such as vasculitis-like rash and vesiculobullous exanthema, are caused by CV-A6 and have been described in children and adults in several countries, such as Argentina [16], Brazil [17], China [18], Israel [19], Italy [20], and Japan [21]. Following the introduction of vaccination against enterovirus (EV)-A71 and CV-A16, CV-A6 has become the leading cause of HFMD in children [22]. EVs have a seasonality. In Thailand, the peak of EV infections occurs during the rainy months [22]. EVs are mainly transmitted via the fecal–oral route, from environmental contaminants, or by respiratory droplets [23]. A study showed that the basic reproductive number (*R*_0_) of CV-A6 (5.94) was much higher than that of EV-A71 (5.06) and of CV-A16 (4.84) [24]. 

CV-A6 belongs to the family *Picornaviridae*, genus *Enterovirus*, and species A, which is classified into 25 genotypes according to the phylogenetic clustering of the capsid gene [25]. Among these, 19 genotypes are known to infect humans. Molecular epidemiological studies related to HFMD have shown that CV-A6, CV-A16, and EV-A71 are the dominant genotypes, especially in many Asian countries [26,27,28]. Similar to other EVs, CV-A6 has a non-segmented positive single-stranded RNA genome of approximately 7534 nucleotides (nt), which contains a single open reading frame (ORF). This ORF encodes a large polyprotein, which is cleaved into four structural proteins (VP1–VP4) and seven non-structural proteins (2A–2C and 3A–3D) [29]. Based on VP1 genotyping, CV-A6 can be divided into four genotypes (A, B, C, and D), and genotype D is further subdivided into D1, D2, and D3 [30]. 

Recombination is commonly observed in EVs and is thought to be a crucial evolutionary strategy by which genetic diversity is generated [31,32,33]. Recombination at non-structural regions rarely affects viral fitness and leads to genetic diversity in the viral population [34,35,36]. An increasing number of recombination events have been observed in the CV-A6 genome. Therefore, a better understanding of the genetic diversity of emerging CV-A6 variants is crucial for the development of effective multivalent vaccines against them and treatment strategies.

In Thailand, HFMD epidemics due to CV-A6 were sporadically detected during the 2008–2011 period [22]. In the rainy season of 2012, however, a large outbreak of CV-A6 was reported in Bangkok, which then disseminated to the rest of the country [34]. Children aged 5 years and under comprised over 90% of the HFMD and herpangina cases; over 60% of the affected children were under 3 years of age [34]. A clear 2-year recurrent cyclical pattern has been documented for CV-A6, with epidemics occurring in 2012, 2014, 2016, and 2018 [22]. 

In this study, we investigated the evolutionary history and molecular epidemiological trends of CV-A6 infection by identifying the circulating genotypes and recombinant strains of CV-A6 in Thailand between January 2019 and October 2022. We compared and examined the genetic traits of these newly sequenced CV-A6 variants and their phylogenetic relationships with other reported global strains by performing VP1 sequence analysis. Additionally, we sequenced and characterized the nearly full-length genomes of 20 representative CV-A6 strains circulating in Thailand to elucidate the occurrence of these recombinant strains.

## 2. Materials and Methods

### 2.1. Sample Collection and Detection of EV RNA

All clinical specimens used in this study were collected as part of routine diagnostics services between January 2019 and October 2022. A total of 755 specimens from patients hospitalized with HFMD were examined for EVs in Thailand. Of these, 507 were throat swabs, 186 stool/rectal swabs, 25 cerebrospinal fluid samples, 19 nasal swabs, 15 serum samples, and 3 sputum samples. Viral nucleic acids were extracted from 200 μL of the supernatant using the magLEAD 12gC instrument (Precision System Science, Chiba, Japan) according to the manufacturer’s instructions. Each sample was subjected to real-time reverse transcription PCR (RT-PCR) for pan-enteroviral screening, using primers and probes targeting the 5′ untranslated region (UTR) of the EV genome, as previously described [35]. For EV-A71, CV-A6, and CV-A16 typing, samples were subjected to another real-time RT-PCR assay, with specific primers and probes targeting the VP1 region [35]. To identify EVs other than EV-A71, CV-A6, and CV-A16, nested PCR using CODEHOP degenerate primers was performed, as previously described [36].

### 2.2. Gene Amplification and DNA Sequencing

Semi-nested RT-PCR was carried out, using the primers listed in Appendix A, to amplify the partial VP1 (amplicon length = 657 bp) and 3Dpol (amplicon length = 1200 bp) gene sequences. The full-length genome of the 20 CV-A6 variants was sequenced using primer sets (listed in Appendix A) designed based on nt sequences available in the GenBank database. Briefly, RT-PCR was conducted in a total volume of 25 μL containing 2–3 μL of 100 ng to 1 µg of total RNA, 0.5 μM of each primer, 12.5 μL of 2X Reaction Mix (containing 0.4 mM of each dNTP and 3.2 mM MgSO_4_), 1 μL of SSIII RT/Platinum Taq Mix, and nuclease-free water. The reverse transcription and first-round PCR were performed using the Superscript III One-Step RT-PCR system with Platinum Taq High Fidelity according to the manufacturer’s instructions (Invitrogen, Carlsbad, CA, USA). The following conditions were used: 45 °C for 40 min, then 40 cycles of 94 °C (30 s), 50 °C (30 s), and 68 °C (1 min 45 s), and a final extension at 68 °C for 5 min. Next, 1 μL of the first-round reaction was used as a template in the second-round PCR, containing second-round primers and PerfectTaq MasterMix (5 PRIME, Darmstadt, Germany), which was performed according to the manufacturer’s instructions. We then performed 40 cycles under the following conditions: 94 °C for 30 s, 50 °C for 30 s, and 72 °C for 90 s, followed by a final extension step at 72 °C for 5 min. Sequencing and product amplification were carried out simultaneously in both forward and reverse directions at the First BASE Laboratories Sdn Bhd (Selangor Darul Ehsan, Malaysia).

### 2.3. Phylogenetic Analysis and Evolutionary Dynamics

Sequencher v.5.1 (Gene Codes Corporation, Ann Arbor, MI, USA) was used for nt sequence assembly. Genome sequences were aligned using CLUSTAL W on the European Bioinformatics Institute (EBI) web server [37]. The MEGA program (v10) was used for phylogenetic tree construction using the maximum-likelihood method, and the best-fit model of nt substitution for each sequence dataset was determined using a correction value in the model selection procedure. For the analyses, the Kimura two-parameter (K2P) model with invariant sites (I) and gamma distribution (Г) parameters was selected as the substitution model for VP1 (K2P with Г was used for the 5´-UTR). The general time reversible (GTR) model with invariant sites (I) and Г distribution was used as the model for the 3Dpol, P1, P2, and P3 regions. During phylogenetic tree construction, bootstrap support values were calculated using 1000 replicates.

To reconstruct the evolutionary history and estimate the nt substitution rate per site per year, as well as the time to the most recent ancestor of CV-A6, Bayesian inference via a Markov chain Monte Carlo (MCMC) framework was implemented in BEAST 2, as previously described [37,38]. We used an uncorrelated relaxed clock model with log-normal distribution for the molecular clock model. To ensure adequate mixing of model parameters, MCMC chains were run for 2 × 10^8^ steps, with sampling every 20,000 simulations from the posterior distribution. Tracer version 1.7.1 (http://tree.bio.ed.ac.uk/software/tracer/ (accessed on 9 November 2022)) was used to check the convergence and effective sample size (ESS) of each estimate. All parameters for the run showed ESS > 200. TreeAnnotator (http://beast.bio.ed.ac.uk/treeannotator (accessed on 9 November 2022)) was used to summarize posterior tree distributions and annotate the estimated values to construct a maximum clade credibility tree after a 10% burn-in. Trees were visualized and annotated in FigTree (http://tree.bio.ed.ac.uk/sofware/figtree/ (accessed on 9 November 2022)).

### 2.4. Recombination Analysis

Recombination was detected using Recombination Detection Program version 4 (RDP4) with seven different algorithms (RDP, GENECONV, BootScan, MaxChi, Chimaera, SiScan, and 3Seq). Potential recombination events were also analyzed using BootScan within RDP4, with a sliding window of 200 nt. A pairwise sequence similarity plot was generated using SimPlot v3.5.1 software [39], with a 200 nt sliding window (moving in 20 nt steps).

### 2.5. Nucleotide Sequence Accession Numbers

The GenBank accession numbers for all newly generated sequences obtained in this study are as follows: OP882311 to OP882484 (partial VP1 gene), OP896539 to OP896712 (partial 3Dpol gene), and OP896713 to OP896732 (complete CV-A6 genome).

## 3. Results

### 3.1. EV detection in Thailand

The patients (N = 755) were aged between 10 days and 32 years old (median 3 years), with a male:female ratio of 1.3:1. The overall percentage of EV positivity was 48.7% (368/755). The results revealed that 23.7% (179/755) of the patients were positive for CV-A6, 3.7% (28/755) for EV-A71, 3.4% (26/755) for CV-A16, and 17.9% (135/755) for other EV types (Figure 1). Excluding CV-A6 and CV-A16, the most common CV genotypes were CV-A4, 6.0% (45/755); CV-A2, 2.5% (19/755); CV-A10, 2.4% (18/755); CV-B3, 0.4% (3/755); CV-B4, 0.3% (2/755); and CV-A5, 0.3% (2/755). Eleven strains were identified as echoviruses (E11, n = 6; E30, n = 3; E16, n = 1; and E21, n = 1).

The frequencies of EV infection varied substantially over the study period (Figure 1), with the highest number occurring in the rainy season (July, August, and September). Prior to the start of the coronavirus disease 2019 (COVID-19) pandemic, 12 different EV genotypes accounted for the 165 identified EV-positive samples; 150 samples (90.9%) were typed as EV species A (CV-A2, A4, A5, A6, A10, A16, and EV-A71) and 15 (9.1%) as species B (CV-B3, B4, E11, E21, and E30). Among these, CV-A4 (27.3%) was the most prevalent among the endemic patients, followed by CV-A6 (16.4%), and EV-A71 (14.5%). EV-positive cases have been sporadic since the ongoing global pandemic of COVID-19. From 2020 to 2021, CV-A6 (44.4%) was the most frequently detected virus among patients with HFMD; this was followed by CV-A16 and other minority species A EVs. One patient (3.7%) was positive for a species B EV (E16). With the decline in COVID-19 cases, the predominant EV genotype detected between January 2022 and October 2022 was CV-A6 (99.3%). One of the patients was a 2-year-old boy from Bangkok infected with EV-A71.

### 3.2. Phylogenetic Analysis

Over the past decade, an increase in the number of CV-A6 recombinant mutants has been reported [40,41]. To find evidence of recombination within the CV-A6 variants characterized in Thailand (CV-A6/TH strains) and the previously defined strains, 174 samples were analyzed using sequencing of their VP1 and 3Dpol regions. The sequences in the VP1 region were assigned to clades A–D, and clade D was further divided into subclades D1–D3 according to recent classification criteria [42]. The phylogenetic tree constructed using the partial VP1 gene is shown in Figure 2A. Molecular characterization based on the partial VP1 sequence of CV-A6-positive samples showed that all the CV-A6/TH strains collected from patients between 2019 and 2022 belonged to the previously reported subclade D3. CV-A6/TH strains shared 82.5–84.8% nt identity with the first identified strain ‘Gdula’ (AY421764). The VP1 region of CV-A6/TH strains shared 92.1–100% nt identities with each other. Widespread clustering showed the co-circulation of the CV-A6/TH strains between 2019 and 2022. The majority (n = 160) of the CV-A6/TH strains analyzed in the study belonged to the D3.1 subclade, while smaller numbers belonged to subclades D3.2 (n = 12) and D3.5 (n = 2). Subclade D3.2, comprising 12 Thai strains, was most closely related to the CV-A6 strain identified in Australia (93.0–94.7% nt identity), France (93.8–94.1% nt identity), India (92.3–93.8% nt identity), and Venezuela (92.8–98.7% nt identity).

Phylogenetic analysis revealed that the nt sequences from the 3Dpol region belonged to 23 bootstrap-supported clades (recombinant forms (RFs); A–X) in agreement with previous reports (Figure 2B) [4,5,6,7,8,9,10,11,12,13,14,15,16,17,18,19,20,21,22,23,24,25,26,27,28,29,30,31,32,33,34,35,36,37,38,39,40,43,44,45,46,47]. Most of the CV-A6/TH strains (84%, 147/174) in this study belonged to the previously described RF-A, while only ~6% (11/174) of the strains were assigned to RF-N (6%), and one strain was classified as RF-H (0.6%). Notably, 15 of the CV-A6/TH strains were assigned to the new RF-Y clade and represented 50% (15/30) of the positive EV samples collected during the 2019–2020 period. 

To gain new insights into the CV-A6 genetic diversity, we determined the nearly complete genome sequences (N = 20) of the CV-A6/TH strains collected between 2019 and 2022 and compared them against reference sequences obtained from GenBank (Figure 3). The nearly complete genome of the CV-A6/TH strains in this study consisted of 7170–7238 nt, which contained a portion of the 5′-UTR and an ORF encoding a polyprotein precursor of 2201 amino acids. All 20 CV-A6/TH strains clustered together and formed a clade supported by a high (96–100%) bootstrap value in the 5′-UTR, P1 region, and P2 region. The highest identity (97.7–99.5% nt identity) within the structural protein of the study strains was to a strain isolated between 2017 and 2018 in China, which belonged to subclade D3 (Table 1). The 3Dpol region of four of the CV-A6/TH strains in this study (CU3494, CU3496, CU3514, and CU3547) showed that these strains belonged to the new RF-Y clade.

### 3.3. Evaluation of the Evolutionary History of CV-A6 Using the VP1 Sequence Dataset

The history of CV-A6 outbreaks in Thailand was reconstructed by performing a phylogenetic analysis of the VP1 sequence samples obtained in this study and previously published sequences. Figure 4 shows the evolutionary relationships between the CV-A6/TH strains and the dates when these strains emerged in Thailand. The Bayesian time-scaled phylogenetic analysis estimated an average rate of evolution over the genome of 4.71 × 10^−3^ substitutions per site per year, with a 95% highest density interval (HDI) of 4.04 × 10^−3^ to 5.46 × 10^−3^. Phylogenetic analysis showed the dynamic nature of the epidemic in Thailand and molecular changes in the CV-A6 VP1 sequences. Seven distinct phylogenetic subclades (labeled D3.1–D3.7) were identified, consistent with the topological structure of the maximum-likelihood (ML) phylogenetic tree. We found that the major study sequences, most of which originated from Taiwan, clustered in subclade D3.1. The estimated mean time to the most recent common ancestor (tMRCA) for subclades D3.1 and D3.2 was similar (13 years). Moreover, the two recombinant forms (RFs; RF-A and RF-Y) emerged within subclade D3.1. We found that the recombination event that generated the RF-Y variants identified in this study occurred 4.8 years ago. Meanwhile, the study samples belonging to RF-H and RF-N clustered in subclade D3.2. It is likely that the RF-N variants in this study that appeared in 2022 underwent recombination between 2016 and 2022. Furthermore, we found that two of the study samples belonged to the same subclade (D3.5) as the published CV-A6/TH sequences identified in 2014–2015, including strains from Japan, Spain, and Germany.

### 3.4. Analysis of Mutation Profiles in CV-A6 Recombinant Subclade D3/Y Genomes

To identify potential recombination events in specific genomic regions of the CV-A6 variants identified in this study, we performed similarity plot and bootscan analyses. Figure 5B shows a similarity plot of the complete EV species A genomic sequences, in which the subclade D3/Y strain identified in this study was used as the query sequence. The SimPlot graph revealed the nt similarity within genomic regions P1, P2, and the beginning of P3, suggesting that these regions are almost identical to those of the earlier CV-A6 strains. The subclade D3/Y strains displayed a relatively high degree of similarity in the 3Dpol protein compared to the CV-A10 strain. Bootscan analysis showed evidence of recombination between the non-structural regions of the CV-A6 and CV-A10 strains (Figure 5C); one breakpoint was identified at nt position 6397. 

To elucidate the mutations in the subclade D3/Y strains from Thailand, we compared the amino acid changes between the four viral strains and the subclade D3/A strains (all from Thailand) characterized in this study and the global reference strains from the NCBI server (Figure 5D). We found that the structural proteins (at 5 positions) contained fewer non-synonymous mutations than the non-structural proteins (at 26 positions). The VP1-encoding gene of EVs plays an essential role in evading the host’s immune response. We found 27 types of synonymous and 3 types of non-synonymous mutations in the VP1 gene of the subclade D3/Y strains. Among them, S58N, V174I, and V299A were present at high frequencies in the D3/Y subclade. We identified 47 types of synonymous and 2 types of non-synonymous mutations in the VP2–VP4-encoding genes. Among them, A142T and R220K were located within the VP3-encoding gene. For the 2A–2C-encoding genes, 52 types of synonymous and 5 types of non-synonymous mutations were identified. Substitutions at five positions (3, 15, and 25 in 2A and 21 and 307 in 2C) were identified in this study. For the non-structural (3A–3Dpol-encoding) genes, 185 types of synonymous and 21 types of non-synonymous mutations were identified. The subclade D3/Y strains were separated from the neighboring lineages by a relatively long branch with 16 amino acid substitutions (3Dpol-S37N, T134I, R136K, K169R, D213N, G259S, N260E, I263V, T299S, S308A, S342L, R346K, E370D, E/K383D, A436T, and I442V).

## 4. Discussion

In this study, we examined the molecular epidemiology of EVs and characterized the CV-A6 genome in Thai children requiring hospital care for hand, foot, and mouth disease and herpangina from January 2019 to October 2022. Over the past decade, sporadically occurring CV-A6 infections have signaled the potential global re-emergence of EV pathogens [1,2,3,4,5,6]. In this study, CV-A6 was the predominant genotype of EVs identified from patient samples, followed by EV-A71 and CV-A16. Our results suggested that children under the age of 5 years are preferentially infected by EVs associated with HFMD. Previous studies also reported a similar correlation between age and the frequency of EV infections [48,49]. In this study, we observed a large increase in HFMD outbreaks during the rainy season in Thailand, with a peak of infections between July and September. Previous studies have shown that sporadic cases of HFMD were caused by CV-A6 in Thailand between 2008 and 2012, when CV-A6 became the main genotype and other EVs were detected sporadically [22,45]. Since then, CV-A6-infected patients have been more frequently identified from 2013 to 2018 [22].

A previous study of CV-A6 epidemiology revealed that the shift from CV-A6 clade D2 to D3 occurred in China and France [40,41]. Moreover, since 2010, CV-A6/D3 strains have caused large HFMD outbreaks in several countries, including Brazil [7], China [8], India [11], Japan [12], and Vietnam [15]. Our study also showed that the CV-A6/D3 strains detected between 2019 and 2022 were phylogenetically closer to viruses circulating in Australia, China, Japan, France, Denmark, and Vietnam between 2013 and 2021, which suggests long-range viral circulation. Molecular comparisons of nearly entire CV-A6 genomes in this study showed that amino acid residues in the structural proteins of the RF-Y clade are almost identical to those of RF-A; the amino acid variation at five positions in the VP3 and VP1 capsid proteins were the only exception. 

In agreement with the results of several previous studies [40,41,47], we found that recombinant subclades existed in the predominant CV-A6 genotype circulating in Thailand. A 2014 Scottish study identified a CV-A6 variant, termed “subclade D3/RF-H”, which was associated with atypical HFMD [47]. Several different morphological patterns, such as widespread vesiculobullous and erosive eruption, eczema coxsackium, and bullous palmoplantar eruption, have been associated with CV-A6 infection [49,50,50]. A previous study has shown that RF-F was the second most frequently occurring RF found in Thailand in 2014 [46]. According to 3Dpol sequencing results, the strains characterized in our study clustered within three previously assigned RFs: RF-A, RF-H, and RF-N. Moreover, most of the CV-A6 variants associated with HFMD in Thailand in 2019 were clustered in a newly identified RF-Y clade, which is distinct from other clades previously characterized. These findings indicate that dramatic changes in the recombination group frequencies coincided with peaks in the CV-A6 5-year incidence cycle. This study also suggests that intertypic recombination events occurred between CV-A6 and other species A EVs. In this study, a new recombinant variant (RF-Y) emerged via recombination within the non-structural region of CV-A10. This is consistent with a previous study from mainland China showing that RF-J and RF-L affect the non-structural region and share ~90% nt sequence homology with three distinct EV-A genomes: CV-A4, CV-A8, and EV-A71 [40]. A study from Hong Kong reported that recombination between the non-structural regions (2C and 3Dpol) of CV-A6 and CV-A4 has risen to EV-A71 [10]. In this study, we presented the first instance of an intertypic recombination event occurring between CV-A6 and CV-A10. This event led to the formation of new recombinant variants and may highlight the evolutionary dynamics and diversity of CV-A6 strains.

In summary, this study broadened our understanding of the epidemiology of CV-A6 infection in Thailand, which will likely have a profound impact on the prevention and control of HFMD. Following its emergence in 2019, the CV-A6/D3 variant continues to dominate EV epidemics in Thailand. Therefore, constant surveillance of CV-A6-associated HFMD will be essential for monitoring how novel CV strains emerge. The increasing number of RFs among CV-A6 clade D3 also highlights the urgent need for future research on efficient vaccines to help eradicate CV-A6 infection.

## Figures and Tables

**Figure 1 viruses-15-00073-f001:**
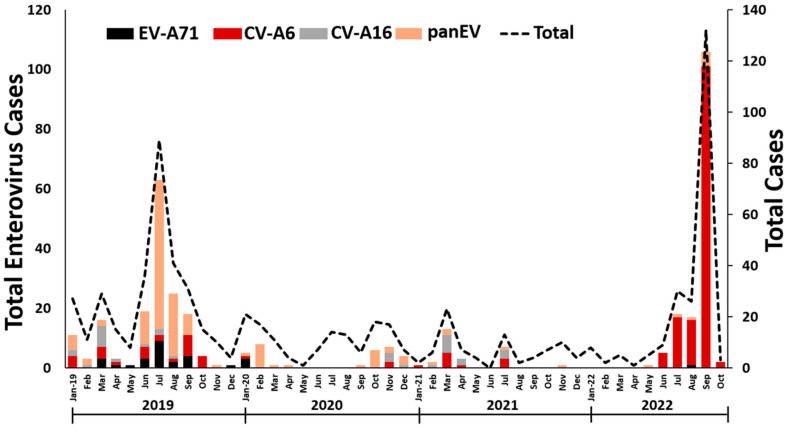
Temporal distribution of enteroviral infections, by epidemiological month, Thailand, 2019–2022 (n = 755).

**Figure 2 viruses-15-00073-f002:**
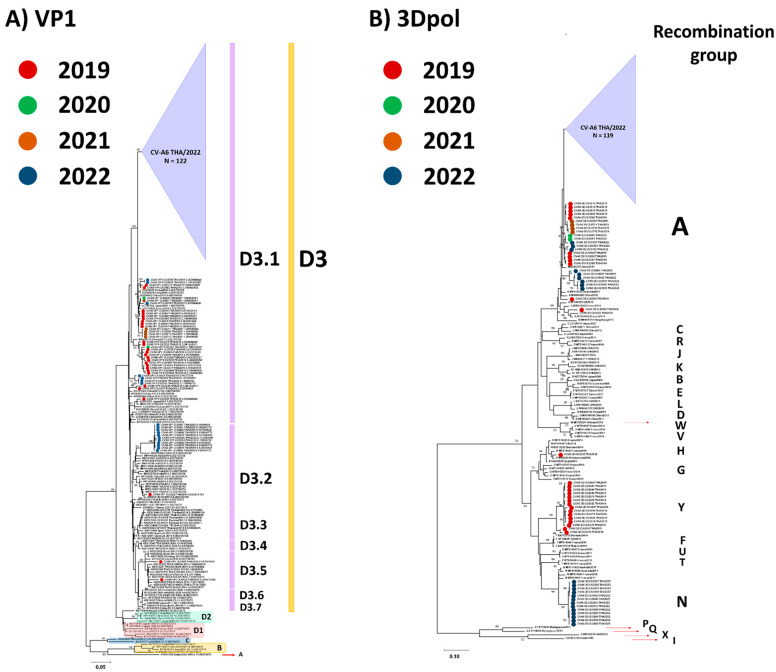
Phylogenetic comparisons of sequences from partial VP1 and 3Dpol. Phylogenetic comparisons of sequences from (**A**) partial VP1 (nt positions 2630–3286, 657 bp) and (**B**) partial 3Dpol (nt positions 6188–7388, 1200 bp) using samples and sequences of other coxsackievirus (CV)-A6 variants obtained from GenBank. Maximum-likelihood trees were reconstructed using 1000 bootstrap re-samples to demonstrate the robustness of groupings; values ≥ 70% are shown.

**Figure 3 viruses-15-00073-f003:**
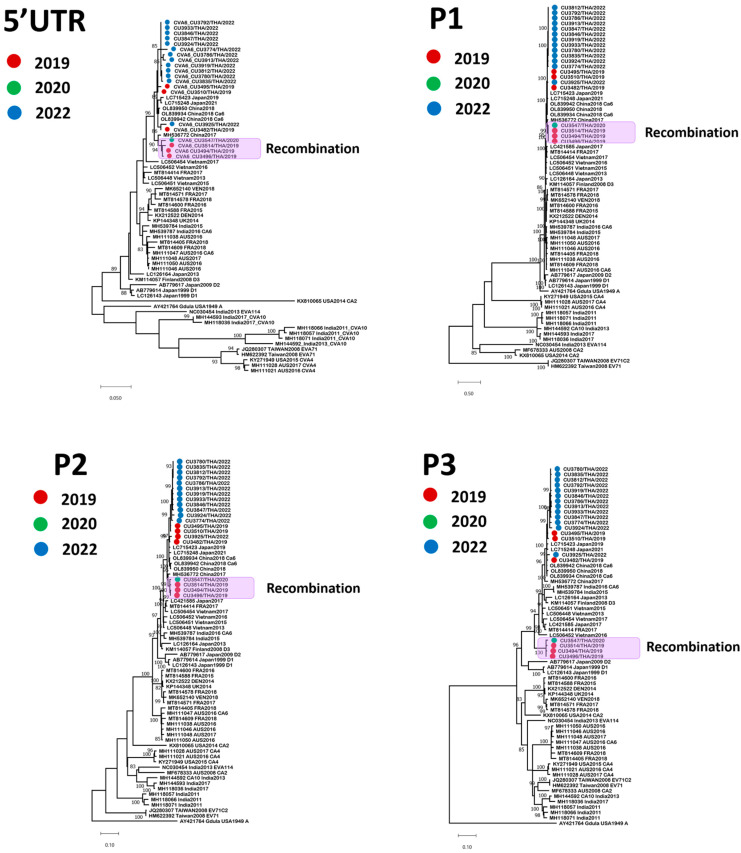
Unrooted phylogenetic trees of coxsackievirus (CV)-A6 strains based on different genomic regions. Maximum-likelihood trees were constructed based on the 5′-UTR and the P1, P2, and 3Dpol regions. Sequences are numbered according to the numbering of the Gdula prototype strain (GenBank accession number AY421764) and previously referenced sequences. Bootstrap values at key nodes are shown as percentages of 1000 replicates. The nearly complete genome sequences of the 20 CV-A6/TH strains, obtained in this study, are highlighted using colored dots. The scale bars represent the number of substitutions per site.

**Figure 4 viruses-15-00073-f004:**
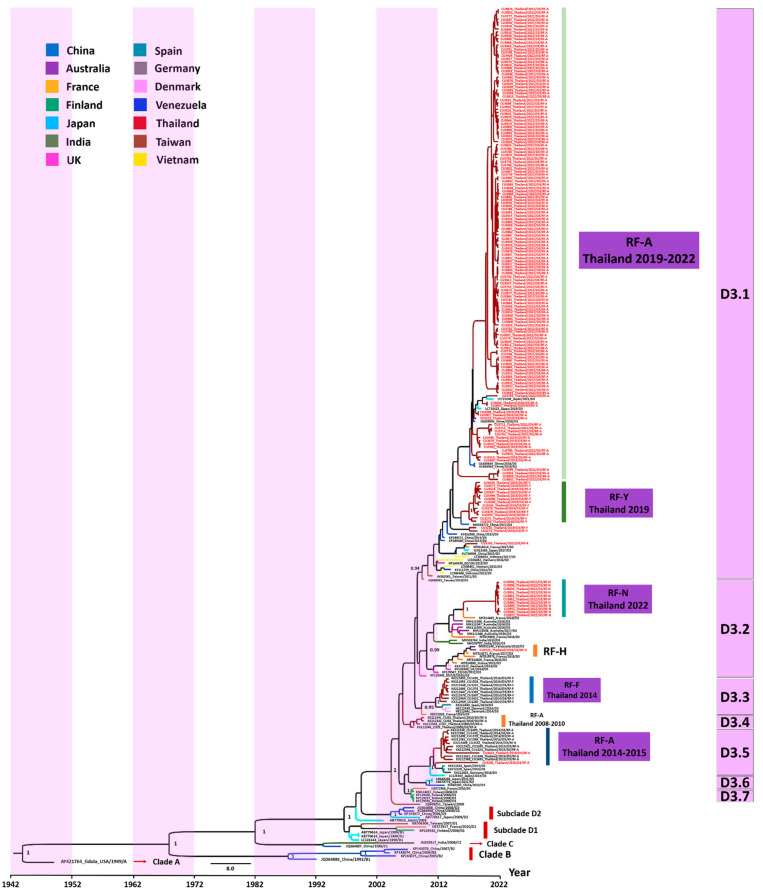
Time-scaled phylogenetic tree of VP1 sequences of coxsackievirus (CV)-A6 variants. The phylogenetic tree was generated using CV-A6 VP1 sequences characterized in this study (red) and previously published sequences. A maximum clade credibility tree was constructed from 10,000 trees, sampled from the posterior distribution with mean node ages. Clades described in previous studies (A, B, C, D1, D2, and D3) are shown. Several subclades were associated with severe hand, foot, and mouth outbreaks in Thailand; posterior probability support is provided. Branch colors denote different countries.

**Figure 5 viruses-15-00073-f005:**
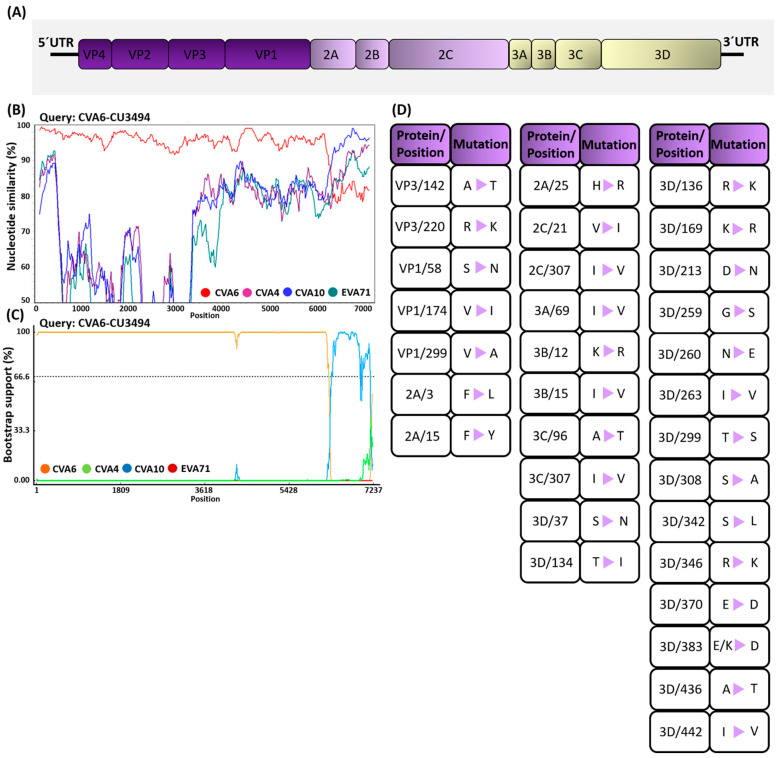
(**A**) Schematic diagram of the enteroviral (EV) genome structure. (**B**) Recombination analysis of putative recombinant coxsackievirus (CV)-A6 strains. SimPlot analysis of the recombinant strain CVA6-CU3494. The vertical axis represents the nucleotide sequence similarity (%) between the putative recombinant strain and other EV-A lineages, constructed using SimPlot v3.5.1 with a window size of 200 nt and a step size of 20 nt. (**C**) Bootscan plot of recombination events between the daughter strain EVA71-CU3494 and major (CV-A6) or minor (CV-A10) parental strains. The bootstrapping support value was computed using the RDP4 program with a window size of 200 nt, a step size of 10 nt, and 1000 bootstrap replicates. (**D**) Amino acid changes between the CV-A6 subclade and the D3/A and D3/Y strains.

**Table 1 viruses-15-00073-t001:** Demographic characteristics and characterization of 20 CV-A6/TH strains based on structural sequences.

CV-A6/TH Strains	Age (years)	Sex	Sample	Collection Date	Subclade	Nearest CV-A6 Strain in Structural Region (Accession No.)	Similarity with the Nearest CV-A6 Strain (% nt)	GenBank Accession No.
CU3482	1	M	Stool	9-Sep-19	D3/A	China/2018 (MN845848)	99.5	OP896719
CU3494	6	F	Throat swab	18-Sep-19	D3/Y	China/2017 (MN845834)	98.5	OP896715
CU3495	3	F	Throat swab	21-Sep-19	D3/A	China/2018 (OL839942)	99.2	OP896717
CU3496	3	M	Throat swab	23-Sep-19	D3/Y	China/2017 (MN845834)	98.4	OP896716
CU3510	8	M	Throat swab	8-Oct-19	D3/A	China/2018 (OL839942)	99.2	OP896718
CU3514	1	M	Throat swab	18-Oct-19	D3/Y	China/2017 (MN845834)	98.6	OP896714
CU3547	3	M	Stool	24-Jan-20	D3/Y	China/2017 (MN845834)	98.3	OP896713
CU3774	1	M	Stool	22-Jun-22	D3/A	China/2018 (OL839942)	98.0	OP896722
CU3780	3	M	Throat swab	1-Jul-22	D3/A	China/2018 (OL839942)	98.0	OP896725
CU3786	4	F	Throat swab	1-Jul-22	D3/A	China/2018 (OL839942)	98.0	OP896729
CU3792	1	M	Nasal swab	5-Jul-22	D3/A	China/2018 (OL839942)	98.0	OP896728
CU3812	2	M	Throat swab	1-Aug-22	D3/A	China/2018 (OL839942)	98.0	OP896726
CU3835	3	M	Throat swab	31-Aug-22	D3/A	China/2018 (OL839942)	97.9	OP896727
CU3846	2	M	Throat swab	8-Sep-22	D3/A	China/2018 (OL839942)	97.9	OP896724
CU3847	3	F	Throat swab	8-Sep-22	D3/A	China/2018 (OL839942)	97.7	OP896723
CU3913	3	F	Throat swab	20-Sep-22	D3/A	China/2018 (OL839942)	97.9	OP896730
CU3919	6	M	Throat swab	20-Sep-22	D3/A	China/2018 (OL839942)	97.7	OP896731
CU3924	4	M	Throat swab	21-Sep-22	D3/A	China/2018 (OL839942)	97.7	OP896721
CU3925	1	M	Throat swab	21-Sep-22	D3/A	China/2018 (MN845848)	98.5	OP896720
CU3933	4	M	Throat swab	21-Sep-22	D3/A	China/2018 (OL839942)	97.7	OP896732

## Data Availability

The datasets generated during this study are available in GenBank (https://www.ncbi.nlm.gov/genome) under accession numbers OP882311 to OP882484 (partial VP1 gene), OP896539 to OP896712 (partial 3Dpol gene), and OP896713 to OP896732 (complete CV-A6 genome).

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
