# Peer review of "Evolutionary and Genetic Recombination Analyses of Coxsackievirus A6 Variants Associated with Hand, Foot, and Mouth Disease Outbreaks in Thailand between 2019 and 2022"

_viruses, 2022, doi:10.3390/v15010073_

Round 1

Reviewer 1 Report

This article examines the molecular epidemiology of hand, foot, and mouth disease (HFMD) isolates of coxsackievirus A6 (CVA6) in Thailand during 2019-2022. The authors demonstrate that in nearly half of cases and enterovirus could be isolated from patient samples and nearly half of these enterovirus isolates were CVA6 but only about 15% were enterovirus A71 (EV-A71) or coxsackievirus A16 (CVA16) which had previously been major isolates in Asia for HFMD. The authors demonstrated that all these CVA6 isolates belonged to the D3 clade (as defined by partial VP1 sequence) and most of the isolates were recombinant form A (RF-A) based on the 3Dpol region. In the period from 2019-2021 the majority of enteroviruses isolated from HFMD cases were CVA4, EV-A71 and CVA16 but during 2022 there was an exceptionally large increase in CVA6 isolates from HFMD.  Examination of the VP1 sequence of isolates by year and by clade demonstrated that while the majority of CVA6 isolates in 2022 belonged to the clade D3.1, a minority population belonged to clade D3.2 and 2019 isolates were from D3.1, D3.2, and D3.5. Complete sequences of 20 of the isolates demonstrated the highest identity of the capsid protein to Chinese isolates of CVA6 2017-2018. The majority of 2022 isolates were RF-A, although a minority were RF-N, and 2019 isolates were RF-A or RF-Y. The authors provided evidence that the D3/RFY strains were recombinant between CVA6 and CVA10 with one breakpoint in the coding region for 3Dpol indicating that the new 2019 RF-Y were generated by a recent intertypic recombination.

Minor issues:

Lines 41-42, the authors should provide a reference for the replacement of EV-A7 and CVA16 in HFMD by CVA6. Probably references 22 and 36.

Lines 59-65: “Recombination at nonstructural regions rarely affects viral fitness”

It is interesting that the authors cite a study of 2A in EV71 affecting virulence when a much more thorough review of such sites is available in Ang et al (Ang, et al. 2021 Emerging Microbes & Infections, 10:1, 713-724, DOI: 10.1080/22221751.2021.1906754). Ang et al. states "Recombination events appear to drive the fitness and virulence of EV-A71 strains, and lead to the emergence of new EVA-71 strains responsible for major outbreaks." This paper gives several examples of recombinants with altered fitness due to recombination at sites in the nonstructural region.

Line 118: “Clustal Won” should be “Clustal W on”

I note that the Supplementary Materials were not available to me and the low resolution of the lettering in Figures 2, 3 and 4 may have affected my ability to review thoroughly.

This is a good analysis of the emerging role of CVA6 in HFMD with a thorough examination of the role of recombination in the emergence of new strains in Thailand, strains which is likely to be informative for the role and evolution of CVA6 elsewhere in HFMD.

Author Response

Response to Reviewer 1 Comments

Point 1: Lines 41-42: the authors should provide a reference for the replacement of EV-A71 and CVA16 in HFMD by CVA6. Probably references 22 and 36.

Response 1: This has been provided (lines 41-43).

Point 2: Lines 59-65: “Recombination at nonstructural regions rarely affects viral fitness”

It is interesting that the authors cite a study of 2A in EV71 affecting virulence when a much more thorough review of such sites is available in Ang et al (Ang, et al. 2021 Emerging Microbes & Infections, 10:1, 713-724, DOI: 10.1080/22221751.2021.1906754). Ang et al. states "Recombination events appear to drive the fitness and virulence of EV-A71 strains, and lead to the emergence of new EVA-71 strains responsible for major outbreaks." This paper gives several examples of recombinants with altered fitness due to recombination at sites in the nonstructural region.

Response 2: This has been added reference citation 36 on page 2, lines 59-65.

Point 3: Line 118: “Clustal Won” should be “Clustal W on”

Response 3:  This has been revised (line 118).

Point 4: I note that the Supplementary Materials were not available to me and the low resolution of the lettering in Figures 2, 3 and 4 may have affected my ability to review thoroughly.

Response 4:  The figures 2, 3 and 4 have been changed to high resolution.

Reviewer 2 Report

Lines 16-29: An abstract is often presented separately from the article and therefore must be able to stand on its own. For this reason, all abbreviations must be defined at their first mention in the abstract itself.

Line 17: infections (plural)

Line 29: delete "help"

Line 43: infections (plural)

Line 344: "the majority of" replace with "most of" (more concise language would be clearer for your reader)

Line 357: variants and (it’s better to have no comma between these phrases)

Author Response

Response to Reviewer 2 Comments

Point 1: Lines 16-29: An abstract is often presented separately from the article and therefore must be able to stand on its own. For this reason, all abbreviations must be defined at their first mention in the abstract itself.

Response 1: This has been modified (lines 16-29).

Point 2: Line 17: infections (plural)

Response 2: This has been modified (line 17).

Point 3: Line 29: delete "help"

Response 3: This has been changed (line 29).

Point 4: Line 43: infections (plural)

Response 4: This has been changed (line 43).

Point 5: Line 344: "the majority of" replace with "most of" (more concise language would be clearer for your reader)

Response 5: This has been modified (line 344).

Point 6: Line 357: variants and (it’s better to have no comma between these phrases

Response 6: This has been modified (line 357).
